# Short Assessment for People with Human Immunodeficiency Virus (HIV) Aged 50 Years or Older: Essential Tests from Comprehensive Geriatric Assessment

**DOI:** 10.3390/v17070887

**Published:** 2025-06-24

**Authors:** Jordi Puig, Pau Satorra, Ana Martínez, Sandra González, Roberto Güerri-Fernández, Itziar Arrieta-Aldea, Isabel Arnau, Anna Prats, Vira Buhiichyk, Cristian Tebe, Eugenia Negredo

**Affiliations:** 1Fundació Lluita Contra Les Infeccions, Department of Infectious Diseases, Hospital Universitari Germans Trias i Pujol, Germans Trias i Pujol, 08916 Badalona, Spain; amartinezv@scienhub.org (A.M.); sandra.gonzalez@coolifting.com (S.G.); aprats@lluita.org (A.P.); vbuhiichyk@lluita.org (V.B.); enegredo@lluita.org (E.N.); 2Faculty of Medicine, Universitat Autònoma de Barcelona, 08193 Barcelona, Spain; 3NURECARE Research Group, Institut d’Investigació i Hospital Germans Trias i Pujol (IGTP), 08916 Badalona, Spain; 4Biostatistics Support and Research Unit Germans Trias i Pujol Research Institute and Hospital (IGTP), 08916 Badalona, Spain; psatorra@igtp.cat (P.S.); ctebe@igtp.cat (C.T.); 5Hospital del Mar, 08003 Barcelona, Spain; rguerri@psmar.cat (R.G.-F.); iarrieta@psmar.cat (I.A.-A.); iarnau@psmar.cat (I.A.); 6Medicine and Life Sciences Department, (MELIS), University Pompeu Fabra, 08003 Barcelona, Spain; 7Faculty of Medicine, Universitat de Vic—UCC, 08500 Vic, Spain; 8CIBERINFEC, Institute of Health Carlos III (ISCIII), 28029 Madrid, Spain

**Keywords:** HIV, geriatric assessment, aging, geriatric syndromes, psychosocial factors, early management

## Abstract

**Background:** Comprehensive geriatric assessments (CGAs) are necessary to address the needs of people with human immunodeficiency virus infection (PWH) aged ≥ 50 years and ensure that they receive high-quality care. We aimed to identify the most effective tests from an extensive CGA to develop a short CGA. **Methods:** This observational, cross-sectional, and analytical study was conducted in three phases: (1) describing PWH aged ≥ 50 and matched controls; (2) jointly analyzing data to identify the most effective tests from the original CGA and develop a short version; and (3) applying the short CGA separately to both groups. **Results:** The most effective tests—the Lawton scale, SPPB, Barber questionnaire, Pittsburgh Sleep Quality Index, and Cognitive Complaints questionnaire—were used to create a short CGA. It identified abnormalities in 77% of PWH flagged by the full CGA, though 65% with the normal short CGA results had at least one abnormal result in the full version. Most false negatives were due to the excluded Hearing-Dependent Activities scale. **Conclusions:** These findings represent an initial step toward developing a short CGA for an easy and rapid identification of PWH aged ≥ 50, beyond a frailty assessment, who may benefit from early clinical management.

## 1. Introduction

Life expectancy is increasing worldwide, and the population is aging rapidly. According to the World Health Organization, the population aged 60 or over will increase from 1 billion in 2020 to 1.4 billion in 2030 and 2.1 billion in 2050 [1]. In addition, the aging process encompasses a series of changes that, over time, lead to progressive functional decline and increased susceptibility to diseases and death [2,3]. Cardiovascular diseases, diabetes, renal diseases, cancer, neurological disorders, and sense organ diseases are some of the most common age-related conditions [4]. Moreover, the aging population is at heightened risk for geriatric syndromes, such as frailty, falls, and polypharmacy, as well as psychosocial aspects, including social isolation and depression [5,6].

Human immunodeficiency virus (HIV) infection has become a chronic condition due to the success of antiretroviral therapies and, as a result, the life expectancy of people with HIV (PWH) has increased considerably [7]. Consequently, so has the number of older PWH, along with their associated comorbidities [8]. In this regard and as an example, Smit et al. predicted that, in 2030, the proportion of PWH aged ≥ 50 years in The Netherlands will reach 73% and that 84% of PWH will have at least one comorbidity.

Notably, aging seems to be accentuated or accelerated in PWH [9]. On the one hand, several age-related comorbidities are more prevalent in PWH compared to non-HIV-infected individuals [10,11]. On the other hand, premature multimorbidity and other age-related conditions, together with an excess of disability and mortality, has been found in PWH [10,12,13]. Moreover, PWH have a lower health-related quality of life than the general population [14].

Given that the early management of older people can improve their morbidity and mortality [15], the need for the close multidisciplinary management of PWH aged ≥ 50 is becoming increasingly evident.

To comprehensively address the needs of PWH aged ≥ 50 and ensure that they receive high-quality care, it is necessary to assess not only their comorbidities and other clinical features but also their psychological, functional, and social dimensions. Thus, a comprehensive geriatric assessment (CGA) is needed in HIV care [16]. However, the multidisciplinary nature of a CGA requires the involvement of several professionals (e.g., nurses, physicians, geriatricians, psychologists, social workers, and physical therapists) and considerable time, which is usually not feasible in routine clinical practice. Consequently, there is a need for a screening tool for PWH aged ≥ 50 that is easier and faster to use in a daily routine to detect patients that may benefit from a more extensive and specific evaluation without compromising the quality of the assessments, relieving the burden on professionals while reaching a broad sample of patients.

To date, no validated tools exist to conduct a short CGA in PWH, beyond just frailty and functional status. Therefore, this study aimed to identify the most effective tests from an extensive CGA to develop a short CGA for PWH aged ≥ 50.

## 2. Material and Methods

### 2.1. Study Design and Participants

This was an observational, cross-sectional, and analytical study. This study was structured in three phases. The first phase consisted of a description of PWH aged ≥ 50 and controls from the Over50 cohort [17], according to their baseline characteristics and the tests included in the original CGA. The cohort included PWH aged ≥ 50 who consecutively visited the HIV units of [blinded for peer review], from September 2015 to January 2024, and age- and sex-matched controls. The second phase involved the joint analysis of data from PWH and their controls to find the most effective tests from the original CGA and constitute a short CGA. Finally, the third phase encompassed the separate analysis of PWH aged ≥ 50 and their controls according to the results of the short CGA in order to evaluate its discriminatory capacity in both cohorts.

A subset of PWH included in the Over50 cohort during different periods were matched by age (±2 years) and sex with non-HIV-infected individuals as controls. They were randomly selected from a list of people attending a primary care center of the same geographical area, as previously described [17], hospital staff members, or patients’ and staff’s relatives.

All participants signed an informed consent form before being included in the Over50 cohort (this is an observational study that follows more than 500 people living with HIV who are over 50 years old). All data were anonymized and handled according to Data Protection Regulation 2016/679 on data protection and privacy for all individuals within the European Union and the local regulatory framework regarding data protection. The study was conducted in accordance with the Declaration of Helsinki, and the protocol was approved by the Ethics Committee of Hospital Germans Trias i Pujol (Code PI-15-106) on (September 2015) and Hospital del Mar (March 2023).

### 2.2. Data Collection and Measures

All data were retrieved from the Over50 Cohort database, including those obtained through the original CGA. Sociodemographic variables included age, sex, education level, and employment and cohabitation status. Clinical variables included body mass index, systolic and diastolic blood pressure, the number of specialists consulted during the last year, the number of admissions and falls during the last year, the number of women with early menopause, smoking habit, and alcohol consumption. The number of comorbidities and polypharmacy was also recorded. The rest of the clinical variables corresponded to those included in the CGA.

The original CGA was performed by a nurse in a single visit, scheduled on the same day the patient visited the HIV unit for other reasons, if possible. The tests included in the original CGA were as follows: the Barthel Index [18], Lawton Instrumental Activities of Daily Living (IADL) scale [19], and Short Physical Performance Battery (SPPB) [20], used to evaluate the patient’s functional status; Functional Ambulation Categories (FAC) [21] to measure ambulation ability; the Hearing-Dependent Daily Activities (HDDA) scale [22], to assess hearing impairment; Pittsburgh Sleep Quality Index (PSQI) [23], used to determine sleep quality; the Barber questionnaire [24] and Fried criteria [25], used to identify patients at high risk of frailty; the Pfeiffer questionnaire [26] and NEUrocognitive (NEU) Screen [27], used to evaluate cognitive impairment; the European AIDS Clinical Society (EACS) cognitive screening questions [28] and the Cognitive Reserve Questionnaire [29], used to assess cognitive complaints and cognitive reserve, respectively; the Lagro-Janssen Gravity Index, used to assess urinary incontinence [30]; the Geriatric Depression Scale (GDS) [31], used to identify depressive symptoms; and the Mini Nutritional Assessment-Short Form (MNA-SF), used to check the nutritional status and food habits [32].

### 2.3. Statistics

Categorical variables were described as frequencies and percentages of each category, whereas continuous variables were presented as the mean and standard deviation (SD) or as the median and interquartile range (IQR, 25th and 75th percentiles), depending on their distribution.

Principal Component Analyses (PCAs) were conducted to identify a reduced set of tests that capture the majority of the information contained in the full set of assessments included in the original CGA, thereby forming the basis for a short CGA version. PCAs were performed both for continuous and categorical scores, indicating only the number of participants with abnormal test results as previously described [33]. Before performing the PCA, the tests with >35% missing values in the original CGA were excluded. In addition, we reversed the quantitative scores of the Barthel Index, Lawton IADL scale, HDDA scale, SPPB, MNA-SF, and Cognitive Reserve Questionnaire so that a higher score consistently represented a worse condition across all variables. This transformation was applied to standardize the directionality of the scales, facilitating the interpretation of PCA loadings and ensuring coherence in the resulting component structure. Estimated variables factor maps from the PCA were represented, showing the Pearson’s correlation coefficient of each score according to the resulting two principal components. Then, a reduced, new set of tests was selected based on the PCA results and the clinicians’ criteria to constitute the short CGA. After that, participants were clustered into different groups according to the number of tests with abnormal results with the short CGA. Participants without any abnormal results from the short CGA but with ≥1 abnormal result from the original CGA were considered false negatives. In addition, these groups were described in terms of the sociodemographic and clinical characteristics according to HIV infection status (controls or PWH). Furthermore, a description of all the scores by cluster was performed. For descriptive statistics, missing data were handled using complete-case analysis. All analyses were performed with the statistical program R software [34] version 4.3.0.

## 3. Results

### 3.1. Characteristics of Study Participants

A total of 541 individuals were included in the study: 394 PWH and 147 controls.

Table 1 summarizes the main characteristics of all study participants. Briefly, their mean age was nearly 71 and 66 years for controls and PWH, respectively, and about 67% of controls and 76% of PWH were males. Approximately 14% of controls and 33% of PWH lived alone. In addition, about 52% controls and 66% PWH were current or former smokers, and their median number of comorbidities was two for controls and three for PWH.

Regarding PWH, the CD4 nadir could be assessed in 358 patients, who showed a median (IQR) of 189.00 (84.50, 288.00) cells/μL. Additionally, the risk group of PWH could be determined in 364 patients, who belonged to the following key population: men who have sex with men (*n* = 149, 40.93%), heterosexual sex (*n* = 104, 28.57%), intravenous drug user (*n* = 78, 21.43%), and other (*n* = 33, 9.07%).

### 3.2. Identification of Essential Tests from Original CGA

Table 2 shows the scores and/or number of participants with abnormal test results for each test of the original CGA. The HDDA scale showed a high number of abnormal results both in controls (*n* = 55, 39.57%) and PWH (*n* = 195, 52.56%), indicating poor discriminatory power. In addition, FAC, NEU Screen, Lagro-Janssen Gravity Index, Fried criteria, GDS, and Cognitive Reserve Questionnaire tests were excluded before conducting the PCAs due to their high number of missing values since most of these tests were not included in the first versions of the original CGA.

The results of the PCAs of tests according to their numeric scores and the number of participants with abnormal results are shown in Figure 1. After obtaining these results, we excluded the following tests from the initial set: Pfeiffer, the HDDA scale, Barthel, and MNA-SF. The Pfeiffer score was excluded because, on the one hand, it showed a very low weight and was highly correlated with SPPB in the categorical PCA and, on the other hand, because it did not show a high weight and was closely correlated with the Lawton IADL scale in the continuous PCA. The HDDA scale score was excluded due to its very low weights in both PCAs. In the case of the Barthel score, it was excluded owing to its high correlation with the Barber questionnaire in both PCAs and its low weight in the categorical PCA. Finally, we excluded MNA-SF because of the low number of patients with abnormal results. Conversely, the EACS cognitive screening questions, despite being highly correlated with the PSQI in the categorical PCA, were included to have at least one score that may capture the participants’ cognitive dimension.

### 3.3. Short CGA: Combination of Lawton IADL Scale, SPPB, Barber Questionnaire, PSQI, and EACS Cognitive Screening Questions

Using the reduced set of tests selected for the short CGA (i.e., a combination of the Lawton IADL scale, SPPB, Barber questionnaire, PSQI, and EACS cognitive screening questions), 33 (60.00%) controls and 199 (68.62%) PWH had ≥1 abnormal test result (Appendix A). The characteristics of study participants according to the number of abnormal test results from this new set are shown in Appendix A.

As can be seen in Table 3, of all 45 controls and 258 PWH with ≥1 abnormal scores from the original CGA, 33 (73.33%) controls and 199 (77.13%) PWH showed ≥1 abnormal scores from the short CGA. However, of all 22 controls and 91 PWH with normal results using the short CGA, 12 (54.55%) controls and 59 (64.84%) PWH showed ≥1 abnormal scores from the original CGA (i.e., false negatives). These numbers of false negatives represent 21.82% of all controls and 20.34% of all PWH evaluated with the short CGA. Appendix A indicates that the majority of these false negatives were attributable to abnormal scores on the HDDA scale. This test was excluded from the short CGA due to its limited discriminatory power, which was compromised by alterations in most of the underlying hearing assessments.

### 3.4. Description of Clinical Frailty Score (CFS) Based on the Value of the Short Assessment

Appendix A shows that individuals in the control group with non-altered values assessed by the short CGA have CFS values that are all lower than or equal to 3. Most frequent values were 1 (45.45%) and 2 (40.91%). For individuals with one or more altered values, the number of missing CFS values in this group is very high, so a proper comparison cannot be made.

Among individuals in the HIV group with non-altered values, 100% had CFS values of 3 or less. Most frequent values were 2 (40.66%) and 1 (38.46%). Among individuals with one altered value, 16.28% had a CFS value greater than 3. Most frequent values were 2 (32.56%) and 3 (30.23%). The percentage increased for individuals with two or more altered values, with 63.33% having a value higher than three. Most frequent values were 4 (43.33%) and 3 (26.67%).

## 4. Discussion

This study shows the most effective assessments for a comprehensive evaluation of PWH aged ≥ 50, by maximizing time efficiency and moving beyond just frailty and functional status. These assessments, which actively incorporated other domains like psychosocial well-being, cognitive function and geriatric syndromes, included the Lawton IADL scale, SPPB, Barber questionnaire, PSQI, and EACS cognitive screening questions. We used a combination of these assessments to develop a short CGA that gathered the most essential patient data beyond a frailty assessment. Approximately 77% of participants with ≥1 abnormal test results in the original CGA had ≥1 abnormal test results with the short CGA. On the other hand, 21.82% of controls and 20.34% of PWH evaluated with the short CGA were false negatives; most of these false negatives corresponded to abnormal scores on the HDDA scale, which showed a very low discriminatory power.

Several studies have previously analyzed comorbidities, frailty, and other geriatric syndromes in cohorts of PWH aged ≥ 50 years [13,35,36]. However, to date, few studies have described the development of a shorter, less time-consuming version of a CGA. In this regard, Bernaud et al. used a simplified geriatric evaluation for characterizing PWH aged ≥ 75 years and assessing their vulnerability [37]. This simplified geriatric evaluation assessed eight health items (i.e., cognition, mood, mobility, autonomy, pain, nutrition, comorbidity burden, and social support), but the tool required further validation, especially in a younger population (50–75 years) and the authors did not describe how it was developed. Additionally, the Clinic Screening Tool CST-HIV, described by Fuster-Ruiz de Apodaca, is another assessment for routine clinical practice intended to identify health issues with a negative impact on the quality of life of PWH [38]. Nevertheless, this tool is only based on patient-reported outcome measures and is mainly focused on geriatric syndromes. Another saving-time approach was that of Sangarlankarn et al., who evaluated the Veterans Aging Cohort Study Index (VACSI) and Geriatric 8 (G8) as screening tools to identify PWH aged ≥ 50 years, who were unlikely to have an abnormal CGA and could therefore avoid undergoing a CGA [39]. In addition, current EACS guidelines and other recommendations for PWH often focus on tools such as gait speed, the SPPB, or the Frail Scale to assess functional status and frailty. However, frailty and functional limitations represent only part of the broader aging-related vulnerability spectrum. Other key domains—including psychosocial factors, nutritional status, cognitive function, sleep quality and other geriatric syndromes—also play a crucial role in identifying individuals at risk and should not be overlooked. The short CGA described in our study may broaden the options for identifying the most functionally impaired PWH aged ≥ 50 years based on an extensive CGA but using a time-saving approach, more applicable to the daily routine clinical practice of professionals without specific training. Similarly to the approach of Sangarlankarn et al., our short CGA is intended to be used as a fast screening tool and, if abnormal results are found, other assessments may be considered.

It is also important to note that the short CGA does not replace standard clinical evaluations. Elements such as comorbidity reviews, polypharmacy assessments, bone health evaluations (e.g., DEXA), and laboratory/radiological monitoring remain part of routine clinical care and are systematically reviewed in the Over50 cohort. The goal of the short CGA is to simplify the use of validated scales for screening key geriatric domains, rather than to replace clinical judgment or omit essential aspects of care.

In our study, 77% of all the PWH with ≥1 abnormal test results in the original CGA had ≥1 abnormal test results with the short CGA. Nonetheless, approximately 20% of participants were false negatives when assessed by the short CGA. These false negatives using the short CGA could lead to the under-detection of patients who need special management. However, most of these false negatives corresponded to individuals with an abnormal score of the HDDA scale, a test not included in the short CGA due to its low discriminatory power, given that abnormal results were prevalent across the majority of participants. Importantly, further analysis confirmed that these abnormal scores were not associated with other domains assessed in the HDDA, suggesting limited overlap with the core components of the short CGA. Then, the time-saving nature of the short CGA would allow healthcare professionals to administer this assessment to all PWH aged ≥ 50 years—or at least to all with comorbidities—as a screening tool, which is more difficult and time-consuming with an extensive CGA. Thus, the short CGA may help preliminarily identify more vulnerable patients, including those who are frail or pre-frail or those with other age-related conditions and would benefit from a more comprehensive evaluation.

The main limitation of this study lies in the lack of validation of the new set of tests to identify the most vulnerable PWH aged ≥ 50 years. Therefore, the results of our study should be interpreted with caution until validation studies are performed, which we are currently working on. In addition, the data to develop the new set of tests were obtained not only from PWH but also from controls. However, PWH accounted for two-thirds of the sample, and we deemed it more important to work with a larger sample of participants for this initial study phase. Moreover, after the new set of tests was established, we analyzed participants’ data according to their HIV infection status and confirmed that the new set of tests was useful for PWH. Another limitation was the high number of missing data on some assessments, especially in the first versions of the original CGA, which prevented us from including several tests for the PCAs and testing their discriminatory capacity such as the frailty phenotype or others. Future versions of the short CGA could benefit from more validated tools in this population. Nevertheless, despite these limitations, to the best of our knowledge, this is the first study to report the development of a short version of an extensive CGA to identify PLWH aged ≥ 50 who are especially vulnerable. Given the lack of dependency and frailty in our sample of patients, we plan to design a future substudy to analyze the role of short CGA in dependent or frail PWH.

## 5. Conclusions

This study presents the development of a short version of the CGA, beyond a frailty and functional assessment, specifically tailored for PWH aged ≥ 50. The proposed short CGA offers a practical and time-efficient tool that may facilitate the identification of individuals at higher risk of health deterioration who could benefit from early and targeted clinical interventions. By combining statistical selection through Principal Component Analysis with expert clinical judgment, this tool balances analytical robustness with clinical applicability. Its simplicity and ease of use make it suitable for routine practice, even in settings where access to specialized geriatric training is limited.

The proposed short CGA is not intended to be a new frailty metric, nor does it aim to replace established frailty assessment tools. The intention is to offer a pragmatic, multidimensional screening tool that identifies individuals at risk of vulnerability in a broader sense. In addition to functional decline and frailty, which are already addressed by guidelines such as those from the EACS, our approach emphasizes the importance of considering other often under-evaluated domains—such as cognitive impairment, psychosocial aspects, sleep quality, nutritional status, and geriatric syndromes. Therefore, the short CGA expands the scope of initial screening, contributing to a more holistic identification of at-risk older PWH who may benefit from a full or adapted CGA.

Although these results are promising, they represent an initial step in the validation process. Additional studies are needed to confirm the predictive value, reliability, and longitudinal utility of the short CGA across diverse clinical settings and populations.

## Figures and Tables

**Figure 1 viruses-17-00887-f001:**
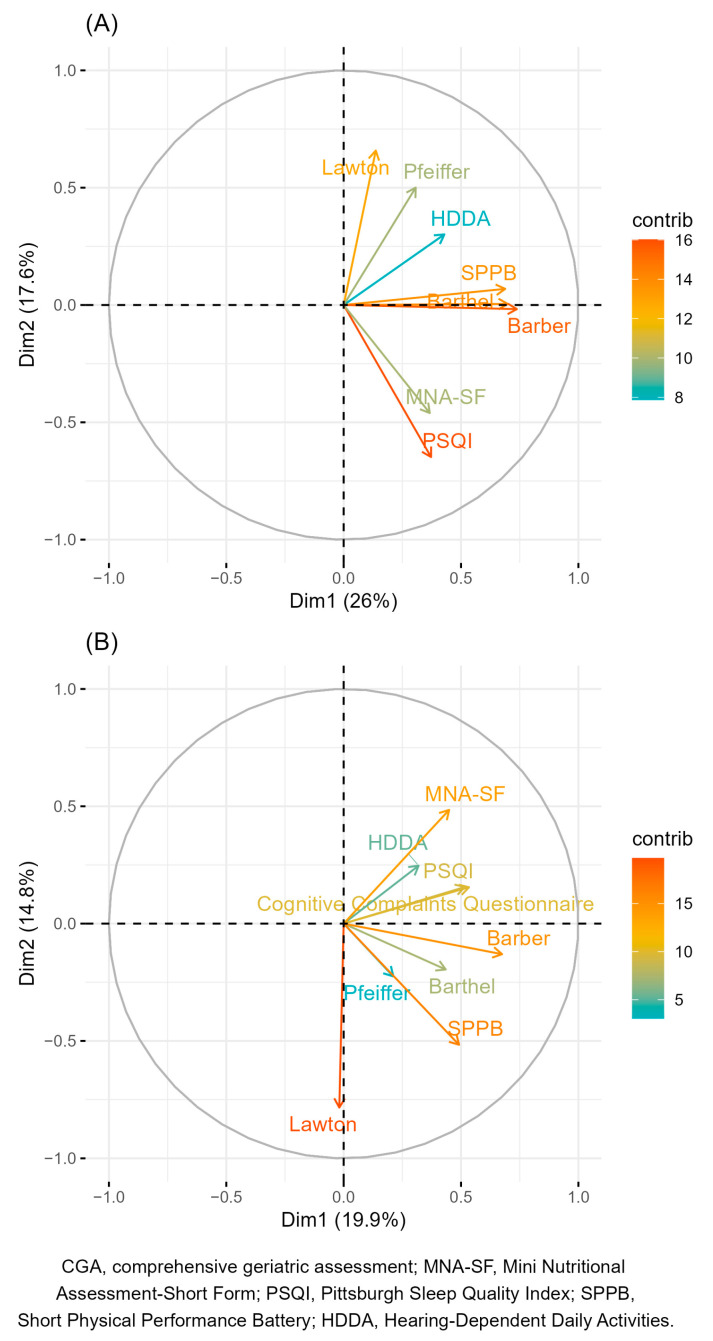
Variable factors map according to the two principal components for numeric scores (**A**) and the number of patients with abnormal test results (**B**) in the original CGA. The color of the arrows indicates the strength of the contribution (in percentage) of each variable to the principal components.

**Table 1 viruses-17-00887-t001:** Characteristics of study population according to HIV infection status.

	Overall (*N* = 541)	*N* Assessed	Control (*N* = 147)	*N* Assessed	PWH (*N* = 394)	*N* Assessed
**Sociodemographic characteristics**						
Age (years), *mean* (*SD*)	67.03 (9.80)	537	70.82 (9.92)	144	65.64 (9.39)	393
Males, *n* (*%*)	396 (73.47)	539	97 (66.90)	145	299 (75.89)	394
Education level, *n* (*%*)		538		146		392
Uneducated	86 (15.99)		50 (34.25)		36 (9.18)	
Primary/secondary education	292 (54.28)		65 (44.52)		227 (57.91)	
Higher education	160 (29.74)		31 (21.23)		129 (32.91)	
Employment status, *n* (*%*)		538		146		392
Unemployed	76 (14.13)		15 (10.27)		61 (15.56)	
Active	294 (54.65)		91 (62.33)		203 (51.79)	
Retired or pensioner	102 (18.96)		11 (7.53)		91 (23.21)	
Other	66 (12.27)		29 (19.86)		37 (9.44)	
Cohabitation status, *n* (*%*)		532		140		392
Living with partner	271 (50.94)		99 (70.71)		172 (43.88)	
Living with a relative	79 (14.85)		18 (12.86)		61 (15.56)	
Living with friends	33 (6.20)		3 (2.14)		30 (7.65)	
Living alone	149 (28.01)		20 (14.29)		129 (32.91)	
**Clinical characteristics**						
Body mass index (kg/m^2^), *median* (*IQR*)	25.52 (23.23, 28.38)	522	27.23 (23.92, 29.40)	140	25.13 (23.02, 27.74)	382
Blood pressure (mmHg), *median* (*IQR*)						
Systolic	135.00 (122.00, 147.00)	473	135.50 (122.00, 148.25)	112	134.00 (122.00, 146.00)	361
Diastolic	80.00 (74.00, 86.00)	474	80.00 (75.00, 89.00)	113	80.00 (74.00, 85.00)	361
Number of specialists consulted during the last year, *n* (*%*)		318		79		239
0	131 (41.19)		44 (55.70)		87 (36.40)	
1	131 (41.19)		31 (39.24)		100 (41.84)	
≥2	56 (17.60)		4 (5.07)		52 (21.76)	
Number of admissions during the last year, *n* (*%*)		475		83		392
0	434 (91.37)		79 (95.18)		355 (90.56)	
≥1	41 (8.63)		4 (4.82)		37 (9.44)	
Number of falls during the last year, *n* (*%*)		522		132		390
0	463 (88.70)		127 (96.21)		336 (86.15)	
≥1	59 (11.31)		5 (3.79)		54 (13.85)	
Early menopause (<45 years old), *n* (*%*)	46 (32.62)	141	21 (45.65)	46	25 (26.32)	95
Smoking habit, *n* (*%*)		527		139		388
Non-smoker	198 (37.57)		67 (48.20)		131 (33.76)	
Current smoker	164 (31.12)		32 (23.02)		132 (34.02)	
Former smoker	165 (31.31)		40 (28.78)		125 (32.22)	
Alcohol consumption (>3 units/day), *n* (*%*)	60 (11.17)	537	7 (4.79)	146	53 (13.55)	391
Number of comorbidities, *median* (*IQR*)	3.00 (1.00, 5.00)	528	2.00 (0.00, 3.00)	141	3.00 (2.00, 5.00)	387
Fecal incontinence, *n* (*%*)	22 (4.25)	518	13 (9.29)	140	9 (2.38)	378
Constipation/defecation urgency, *n* (*%*)	25 (4.84)	517	3 (2.14)	140	22 (5.84)	177

HIV, human immunodeficiency virus; PWH, people with HIV.

**Table 2 viruses-17-00887-t002:** Scores and/or number of patients with abnormal test results for each test included in the original CGA according to HIV infection status.

	Overall (*N* = 541)	*N* Assessed	Control (*N* = 147)	*N* Assessed	PWH (*N* = 394)	*N* Assessed
**Barthel Index**						
Score, *median* (*IQR*)	100.00 (100.00, 100.00)	511	100.00 (100.00, 100.00)	134	100.00 (100.00, 100.00)	377
Abnormal, *n* (*%*)	4 (0.78)	511	0 (0.00)	134	4 (1.06)	377
**Lawton**						
Score, *n* (*%*)		512		136		376
1	7 (1.37)		1 (0.74)		6 (1.60)	
2	16 (3.13)		4 (2.94)		12 (3.19)	
3	208 (40.63)		68 (50.00)		140 (37.23)	
4	4 (0.78)		0 (0.00)		4 (1.06)	
5	235 (45.90)		57 (41.91)		178 (47.34)	
6	2 (0.39)		0 (0.00)		2 (0.53)	
7	2 (0.39)		0 (0.00)		2 (0.53)	
8	38 (7.42)		6 (4.41)		32 (8.51)	
Abnormal, *n* (*%*)	328 (64.19)	511	109 (80.74)	135	219 (58.24)	376
**Hearing-Dependent Daily Activities scale**						
Score, *median* (*IQR*)	22.00 (18.00, 22.00)	510	22.00 (19.00, 22.00)	139	21.00 (18.00, 22.00)	371
Abnormal, *n* (*%*)	250 (49.02)	510	55 (39.57)	139	195 (52.56)	371
**Lagro-Janssen Index**						
Score, *median* (*IQR*)	3.00 (3.00, 3.00)	309	3.00 (3.00, 3.00)	113	3.00 (3.00, 3.00)	196
Abnormal, *n* (*%*)	18 (5.83)	309	2 (1.77)	113	16 (8.16)	196
**Pittsburgh Sleep Quality Index**						
Score, *median* (*IQR*)	7.00 (5.00, 11.00)	446	8.00 (7.00, 10.50)		7.00 (4.00, 11.00)	
Abnormal, *n* (*%*)	26 (5.83)	446	1 (1.49)	67	25 (6.60)	379
**Pfeiffer**						
Score, *n* (*%*)		499		135		364
0	421 (84.37)		120 (88.89)		301 (82.69)	
1	47 (9.42)		9 (6.67)		38 (10.44)	
2	17 (3.41)		1 (0.74)		16 (4.40)	
3	5 (1.00)		3 (2.22)		2 (0.55)	
4	4 (0.80)		0 (0.00)		4 (1.10)	
5	2 (0.40)		2 (1.48)		0 (0.00)	
6	0 (0.00)		0 (0.00)		0 (0.00)	
7	1 (0.20)		0 (0.00)		1 (0.27)	
8	1 (0.20)		0 (0.00)		1 (0.27)	
9	0 (0.00)		0 (0.00)		0 (0.00)	
10	1 (0.20)		0 (0.00)		1 (0.27)	
Abnormal, *n* (*%*)	14 (2.81)	499	5 (3.70)	135	9 (2.47)	364
**Fried**						
Score, *median* (*IQR*)	1.00 (0.00, 2.00)	344	1.00 (0.00, 2.00)	63	1.00 (0.00, 2.00)	281
Abnormal, *n* (*%*)	40 (11.63)	344	5 (7.94)	63	35 (12.46)	281
**Short Physical Performance Battery**						
Score, *median* (*IQR*)	11.00 (9.00, 12.00)	484	11.00 (8.00, 12.00)	135	11.00 (10.00, 12.00)	349
Abnormal, *n* (*%*)	125 (25.83)	484	52 (38.52)	135	73 (20.92)	349
**Mini Nutritional Assessment-Short Form**						
Score, *median* (*IQR*)	13.00 (11.00, 14.00)	482	14.00 (13.00, 14.00)	124	13.00 (11.00, 14.00)	358
Abnormal, *n* (*%*)	127 (26.35)	482	16 (12.90)	124	111 (31.01)	358
**Barber**						
Score, *median* (*IQR*)	0.00 (0.00, 1.00)	463	0.00 (0.00, 0.00)	113	0.00 (0.00, 1.00)	350
Abnormal, *n* (*%*)	61 (13.17)	463	12 (10.62)	113	49 (14.00)	350
**Geriatric Depression Scale**						
Score, *median* (*IQR*)	3.00 (1.00, 6.00)	195	1.00 (0.00, 2.00)	26	3.00 (1.00, 6.00)	169
Abnormal, *n* (*%*)	55 (28.21)	195	2 (7.69)	26	53 (31.36)	169
**Cognitive Reserve Questionnaire**						
Score, *median* (*IQR*)	14.00 (9.00, 16.00)	191	16.00 (10.00, 18.00)	25	13.00 (9.00, 16.00)	166
Abnormal, *n* (*%*)		191		25		166
Lower range	20 (10.47)		0 (0.00)		20 (12.05)	
Low–medium range	30 (15.71)		4 (16.00)		26 (15.66)	
Medium–high range	60 (31.41)		5 (20.00)		55 (33.13)	
Superior range	81 (42.41)		16 (64.00)		65 (39.16)	
**Clinical Frailty Scale**						
Score, *n* (*%*)		198		27		171
Very fit	58 (29.29)		12 (44.44)		46 (26.90)	
Well	64 (32.32)		10 (37.04)		54 (31.58)	
Managing well	45 (22.73)		4 (14.81)		41 (23.98)	
Vulnerable	20 (10.10)		1 (3.70)		19 (11.11)	
Mildly frail	1 (0.51)		0 (0.00)		1 (0.58)	
Moderately frail	8 (4.04)		0 (0.00)		8 (4.68)	
Severely frail	2 (1.01)		0 (0.00)		2 (1.17)	
Very severely frail	0 (0.00)		0 (0.00)		0 (0.00)	
Terminally ill	0 (0.00)		0 (0.00)		0 (0.00)	
**Functional Ambulation Categories**						
Score, *n* (*%*)		197		27		170
0	3 (1.52)		1 (3.70)		2 (1.18)	
1	1 (0.51)		0 (0.00)		1 (0.59)	
2	7 (3.55)		0 (0.00)		7 (4.12)	
3	5 (2.54)		0 (0.00)		5 (2.94)	
4	16 (8.12)		1 (3.70)		15 (8.82)	
5	165 (83.76)		25 (92.59)		140 (82.35)	
Abnormal, *n* (*%*)	11 (5.58)	197	1 (3.70)	27	10 (5.88)	170
**NEUrocognitive Screen**						
Abnormal, *n* (*%*)	47 (24.61)	191	5 (18.52)	27	42 (25.61)	164
**Cognitive Complaints**						
Abnormal, *n* (*%*)	88 (19.05)	462	9 (7.69)	117	79 (22.90)	345

CGA, comprehensive geriatric assessment; HIV, human immunodeficiency virus; PWH, people with HIV.

**Table 3 viruses-17-00887-t003:** Number of tests with abnormal results from original CGA according to HIV infection status and number of abnormal test results from short CGA, ***N***
*(%)*.

Abnormal Results in the Tests Included in the Original CGA	Abnormal Results in the Tests Included in the Short CGA					
	Control			PWH		
	Normal Results (*N* = 22)	1 Abnormal Test Result (*N* = 17)	≥2 Abnormal Test Results (*N* = 16)	Normal Results (*N* = 91)	1 Abnormal Test Result (*N* = 116)	≥2 Abnormal Test Results (*N* = 83)
0	10 (45.45)	0 (0.00)	0 (0.00)	32 (35.16)	0 (0.00)	0 (0.00)
1	9 (40.91)	11 (64.71)	0 (0.00)	35 (38.46)	39 (33.62)	0 (0.00)
2	2 (9.09)	6 (35.29)	8 (50.00)	17 (18.68)	42 (36.21)	14 (16.87)
3	1 (4.55)	0 (0.00)	4 (25.00)	4 (4.40)	22 (18.97)	27 (32.53)
≥4	0 (0.00)	0 (0.00)	4 (25.00)	3 (3.30)	13 (11.21)	42 (50.60)

CGA, comprehensive geriatric assessment; HIV, human immunodeficiency virus; PWH, people with HIV.

## Data Availability

The datasets used and/or analyzed during the current study are available from the corresponding author (JP) on reasonable request.

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
