# Peer review of "Short Assessment for People with Human Immunodeficiency Virus (HIV) Aged 50 Years or Older: Essential Tests from Comprehensive Geriatric Assessment"

_viruses, 2025, doi:10.3390/v17070887_

Round 1

Reviewer 1 Report

Comments and Suggestions for Authors

This analysis assesses the reliability and potential utility of a derived short version of a modified Geriatric Comprehensive Assessment in a cohort of older persons living with HIV (OPLWH). The rationale for this study is due to the increasing prevalence of OPLWH, in both high-income and low-to-middle-income (LMIC) countries. This results in more OPLWH with a greater burden of the premature onset of age-related co-morbidities and geriatric syndromes.  The increased number of such challenging OPLWH will place a predictably increased load on care providers. As not all OPLWH develop the same clinical profile, ie, aging is a heterogenous process, the important question is how to simply and reliably identify those OPLWH most in need of an adapted version of the geriatric model of care which is increasingly being recommended for this especially vulnerable population. Several aspects of this important analysis require clarification in order to understand the clinical implications of the results obtained in this important and potentially useful analysis.

It is relevant to review the fundamental description and purpose of the comprehensive geriatric assessment (CGA). It is a multidimensional and multidisciplinary diagnostic process that evaluates the medical, functional, psychological and social capabilities of a frail older person to develop a coordinated and integrated plan for treatment and long germ follow-up(Pilotto, JAMDA 2017). There are no pre-specified models as to how to constitute the interdisciplinary team and a CGA can occur in any clinical setting. It is also important review the evidence of the effectiveness of a CGA in different settings. As the authors correctly note in the Intro, based on a recent Cochrane Review a CGA has only been shown to improve geriatric relevant outcome parameters in hospitalized elderly patients. The evidence is not very strong that a CGA improves outcomes when performed in an ambulatory setting, for various reasons.

1-The authors state in the Intro that “a CGA is needed in HIV Care”. Do they mean that everyone > 50 should undergo a CGA?

2-Given the realistic organizational and time-limiting realities of performing a complete CGA, the authors also state that “there is a need for a screening tool for PHW>50 that is easier and faster to use in a daily routine to detect pts that may benefit form a more extensive and specific evaluation”. Presumably the more specific evaluation they are referring to is a full CGA.  This is the rationale of this analysis to develop a short CGA as the screening tool. However, recommendations for how to screen for pts who should have a full CGA, both in the general population and in OPLWH, already exist. Screening at-risk pts for frailty in the general population is the recommended 1st step in identifying persons who require a full, or adapted, CGA (Garrard, Aging Clin Expt’l Res 2019). In PWH > 50 the EACS guidelines also recommend screening for frailty using either gait speed, the SPPB or the FRAIL Scale. Pts thus identified as frail are recommended to be referred to an Aging-HIV clinic or other appropriate aging knowledgeable provider. Nevertheless it is a known limitation that there are many frailty metrics and there is no consensus on a single, simple and reliable tool to use.

3-Are the authors implying that their derived short CGA is a new frailty metric? It is difficult to see if that is in fact their recommendation as they actually incorporated the established Frailty Phenotype (FP) model as a parameter into their development of the short CGA. Curiously they did not retain the FP in the final analysis because of too many missing values in their initial assessments. Therefore, one must conclude that their derived shot CGA is not a screen for frailty.

4-Furthermore, it is not clear that the initial assessments of their Cohort were in fact formal CGAs, as they are generally understood. The authors refer to the original description of their Over50 Cohort in a previous publication who underwent a CGA. A careful reading of the Methods in that paper (Negredo, AIDS Care 2022) states that “all pts underwent an intensive multidisciplinary assessment”. However, in the Data Collection section of this submitted paper it is stated that “the original CGA was performed by a nurse in a single visit”. A single care provider evaluating 15 parameters on a single day, without any mention of how that information was then used clinically does not constitute a CGA, as is generally understood. However, an extensive set of clinically relevant assessments were clearly performed.  What is unclear is how this information as then used clinically.

5-Of the numerous originally preformed assessments, the Pfeiffer questionnaire is a screening tool for both delirium and dementia in the elderly which has not been validated in OPLWH. Similarly, the Barber tool is an over 40-year-old mailed-in tool for functional assessment which also has not been validated in OPLWH. More current tools which have been specifically studied in OPLWH are available. In this regard, it is confusing why the Barthel score was excluded because of “its high correlation with the Barber score”. Direct assessment of ADL function by history is a basic geriatric tool. Perhaps the Barber score should have been excluded?

6-It is standard practice to show statistical comparisons between the Control and PWH groups in Table 1, even if there is likely no significant difference between them.

7-In section 3.2 of the Results, as noted above for the exclusion of the FP, it is unclear why results for so many clinically relevant parameters were not recorded during the original assessment. Perhaps it was too difficult for one person to do so many evaluations in a single day? This speaks to the clinical utility of the originally performed CGA and of the reliability of the derived short CGA as being truly representative of the multidisciplinary and multifactorial nature of a CGA.

8-Table 1 also reports results of the Clinical Frailty Scale (CFS), which however is not discussed at all in the paper. This is actually an important and widely recognized frailty assessment tool. The Table shows that almost 90% of the overall OPLWH cohort were either non-frail or vulnerable-to-mildly frail. In general, using this tool, only about 7-8% of the PWH (CFS>5) would have been referred for a full CGA, as per recommendations for non-HIV exposed older adults. This is hardly a burdensome number and is much simpler and quicker to perform than the several time-consuming tests that this paper’s results recommend should be done as per the short CGA.    

9-There is no assessment of polypharmacy (PP). This is a fundamental part of the evaluation of all elderly persons and especially OPLWH, given the known increased risk of PP in OPLWH with possible downstream consequences due to potential drug interactions and toxicities.

10-The high rate of 20% of false negatives obtained with the short CGA is unacceptable. In this regard, it would be relevant to know what the CFS score was for the OPWH who actually were the false-negatives. Would the CFS have identified them? Similarly, what was the FP diagnosis of the false-negatives?

11-There is no doubt that assessing IADLs, lower extremity function with the SPPB, sleep hygiene, and cognition and functional status are all important. What is not clear is how the authors recommend using these 5 parameters. Only 50% of the PWH who had >4 abnormal tests on the original “complete” CGA had more >2 abnormal results on the short CGA.  What do the authors suggest should be minimum criteria to refer for a full CGA?

 12-At the very least a comparison between the short CGA (how many abnormal?)  and an already validated and recommended frailty metric should be considered. The authors have already assessed 2 such metrics, the CFA and the FP (although the FP does take about 10-15 minutes to complete and requires the availability of an expensive hand-grip dynamometer). Determining the CFS takes less than 10 minutes and no equipment.    

Author Response

We would like to thank the reviewers for their detailed and constructive feedback. Below, we provide a point-by-point response to the comments and questions raised.

Comments 1: The authors state in the Intro that “a CGA is needed in HIV Care”. Do they mean that everyone > 50 should undergo a CGA?
Response 1: Thank you for you for pointing this out. We do not propose that all individuals over 50 years with HIV should undergo a full CGA. As stated in the introduction, we recognize that a full CGA is not feasible in daily clinical practice. Therefore, we developed a short version as a screening tool to identify those who might benefit from a more extensive evaluation to detect patients with early frailty or other conditions

Comments 2. Given the realistic organizational and time-limiting realities of performing a complete CGA, the authors also state that “there is a need for a screening tool for PHW>50 that is easier and faster to use in a daily routine to detect pts that may benefit from a more extensive and specific evaluation”. Presumably the more specific evaluation they are referring to is a full CGA. This is the rationale of this analysis to develop a short CGA as the screening tool. However, recommendations for how to screen for pts who should have a full CGA, both in the general population and in OPLWH, already exist. Screening at-risk pts for frailty in the general population is the recommended 1st step in identifying persons who require a full, or adapted, CGA (Garrard, Aging Clin Expt’l Res 2019). In PWH > 50 the EACS guidelines also recommend screening for frailty using either gait speed, the SPPB or the FRAIL Scale. Pts thus identified as frail are recommended to be referred to an Aging-HIV clinic or other appropriate aging knowledgeable provider. Nevertheless it is a known limitation that there are many frailty metrics and there is no consensus on a single, simple and reliable tool to use.
Response 2: We acknowledge that the EACS guidelines and other recommendations suggest the use of tools such as gait speed, the SPPB, or the FRAIL Scale to assess functional status and frailty in PWH >50. However, we believe that frailty and functional limitations represent only part of the broader aging-related vulnerability spectrum. Other key domains—including psychosocial factors, nutritional status, cognitive function, sleep quality, and other geriatric syndromes—also play a crucial role in identifying individuals at risk and should not be overlooked. Our proposed short screening tool aims to provide a more comprehensive yet time-efficient assessment by incorporating several of these often under-assessed dimensions. This approach seeks to complement existing guidelines by broadening the scope of the initial evaluation and potentially improving the detection of individuals who would benefit from a full or adapted CGA. This rationale and perspective have been further detailed in the revised Discussion section of the manuscript.

Comments 3. Are the authors implying that their derived short CGA is a new frailty metric? It is difficult to see if that is in fact their recommendation as they actually incorporated the established Frailty Phenotype (FP) model as a parameter into their development of the short CGA. Curiously they did not retain the FP in the final analysis because of too many missing values in their initial assessments. Therefore, one must conclude that their derived shot CGA is not a screen for frailty.
Response 3: The short CGA is not intended to be a new frailty metric but rather a multidimensional screening tool.Regarding the Fried phenotype, although this model was initially considered, it was excluded from the final analysis due to a high proportion of missing values. However, the short CGA does not aim to replace existing frailty metrics but to serve as an initial step in identifying vulnerability in a broader sense. As we mentioned before, we believe that a comprehensive assessment in older PWH should go beyond frailty and functional status alone. Other domains—such as psychosocial factors, cognitive impairment, geriatric syndromes, and nutritional status—are equally relevant and often under-evaluated in routine clinical practice. Our proposed tool aims to expand the assessment to include these key dimensions in a brief and feasible format.

Comments 4. Furthermore, it is not clear that the initial assessments of their Cohort were in fact formal CGAs, as they are generally understood. The authors refer to the original description of their Over50 Cohort in a previous publication who underwent a CGA. A careful reading of the Methods in that paper (Negredo, AIDS Care 2022) states that “all pts underwent an intensive multidisciplinary assessment”. However, in the Data Collection section of this submitted paper it is stated that “the original CGA was performed by a nurse in a single visit”. A single care provider evaluating 15 parameters on a single day, without any mention of how that information was then used clinically does not constitute a CGA, as is generally understood. However, an extensive set of clinically relevant assessments were clearly performed. What is unclear is how this information as then used clinically.
Response 4: While the assessments were extensive and multidimensional, they were conducted by a single nurse in one visit. Therefore, we prefer to refer to them as 'comprehensive geriatric evaluations' rather than formal CGAs. In addition to the tools (questionnaires, test, index, scales) included for the current analysis, the comprehensive assessment includes a review of comorbidities, polypharmacy, diagnostic tests such as X-rays, ECG and DEXA scans, vaccination status, referrals to specialists like gynecology, and screenings such as anal cancer screening, etc
This limitation is discussed in the manuscript in the Discussion section.

Comments 5. Of the numerous originally preformed assessments, the Pfeiffer questionnaire is a screening tool for both delirium and dementia in the elderly which has not been validated in OPLWH. Similarly, the Barber tool is an over 40-year-old mailed-in tool for functional assessment which also has not been validated in OPLWH. More current tools which have been specifically studied in OPLWH are available. In this regard, it is confusing why the Barthel score was excluded because of “its high correlation with the Barber score”. Direct assessment of ADL function by history is a basic geriatric tool. Perhaps the Barber score should have been excluded?
Response 5: We appreciate this observation. The Barthel was excluded due to its high correlation with the Barber. Although the Barber has not been specifically validated in people with HIV, its inclusion was based on its discriminative capacity in our analysis. We recognize that future versions could benefit from more validated tools in this population. A validation study is planned, and all these recommendations will be taken into account. This information has been included in the manuscript in the Discussion section.

Comments 6. It is standard practice to show statistical comparisons between the Control and PWH groups in Table 1, even if there is likely no significant difference between them.
Response 6: We agree with the reviewer that statistical comparisons in tables like our table 1 is a standard practice. However, the purpose of table 1 was to describe the demographic and clinical characteristics of the included subjects by Control and PWH groups. Performing statistical comparisons by groups would involve more than 15 comparisons, increasing the risk of false-positive findings. In this context, p-values would not provide a reliable answer as to whether the observed differences are meaningful or misleading. The STROBE (Strengthening the Reporting of Observational Studies in Epidemiology) is clear at this point “Inferential measures such as standard errors and confidence intervals should not be used to describe the variability of characteristics, and significance tests should be avoided in descriptive tables”.

Comments 7. In section 3.2 of the Results, as noted above for the exclusion of the FP, it is unclear why results for so many clinically relevant parameters were not recorded during the original assessment. Perhaps it was too difficult for one person to do so many evaluations in a single day? This speaks to the clinical utility of the originally performed CGA and of the reliability of the derived short CGA as being truly representative of the multidisciplinary and multifactorial nature of a CGA.
Response 7: As we mentioned before, in addition to the tools (questionnaires, test, index, scales) included for the current analysis, the comprehensive assessment includes a review of comorbidities, polypharmacy, diagnostic tests such as X-rays, ECG and DEXA scans, vaccination status, referrals to specialists like gynecology, and screenings such as anal cancer screening, etc
Some assessments were not included in the earliest versions of the protocol, which resulted in missing data for certain parameters. This limitation is acknowledged in the manuscript. It is important to note that this cohort was initiated approximately almost ten years ago, at a time when the number of people living with HIV aged >50 was significantly smaller and when comprehensive geriatric assessment was not yet widely implemented or prioritized in this population. The protocol initially included a limited set of tests, and additional domains were progressively incorporated over the years as awareness and clinical practices evolved. Despite these limitations, the components ultimately retained for the short CGA demonstrated good discriminative capacity and are feasible for routine clinical application. This supports its potential utility as a pragmatic and multidimensional screening tool.

Comments 8. Table 1 also reports results of the Clinical Frailty Scale (CFS), which however is not discussed at all in the paper. This is actually an important and widely recognized frailty assessment tool. The Table shows that almost 90% of the overall OPLWH cohort were either non-frail or vulnerable-to-mildly frail. In general, using this tool, only about 7-8% of the PWH (CFS>5) would have been referred for a full CGA, as per recommendations for non-HIV exposed older adults. This is hardly a burdensome number and is much simpler and quicker to perform than the several time-consuming tests that this paper’s results recommend should be done as per the short CGA.
Response 8: We appreciate this observation. We will add a discussion on the CFS in the manuscript. Although it was not included in the short CGA, its simplicity and validation make it a useful tool for future comparison. However, we would point out that the objective of the current study was to identify the most sensitive tools for a comprehensive assessment beyond frailty and functional status alone, by including other domains.

Comments 9. There is no assessment of polypharmacy (PP). This is a fundamental part of the evaluation of all elderly persons and especially OPLWH, given the known increased risk of PP in OPLWH with possible downstream consequences due to potential drug interactions and toxicities.
Response 9: We fully agree with the reviewer that polypharmacy is a fundamental component of the evaluation in older people living with HIV (OPLWH), given the increased risk of drug–drug interactions, adverse events, and treatment burden in this population. However, we would like to clarify that polypharmacy, along with comorbidity review, radiologic evaluations (e.g., X-rays), and bone density assessments (e.g., DEXA scans), has not been excluded from the clinical evaluation of the Over50 cohort participants. These elements continue to be routinely assessed and recorded as part of their comprehensive clinical care. The aim of the present study was not to eliminate these evaluations. The proposal is to maintain all of them. However, our objective is to identify and simplify the set of structured tests or scales (i.e., standardized instruments- questionnaires, test, index, scales) that could constitute a short, time-saving geriatric screening tool. This clarification has been added to the Discussion section of the manuscript.

Comments 10. The high rate of 20% of false negatives obtained with the short CGA is unacceptable. In this regard, it would be relevant to know what the CFS score was for the OPWH who actually were the false-negatives. Would the CFS have identified them? Similarly, what was the FP diagnosis of the false-negatives?
Response 10: This could be a limitation. However, we finally considered it acceptable since, as indicated in the manuscript, most false negatives were due to the exclusion of the HDDA from the short assessment. We decided the exclusion of HDDA from the short version since almost all participants in the original CGA presented abnormal result, which showed low discriminative power.
We will evaluate whether tools like the CFS or FP could have identified these cases and add this comparison in the revision.
For all patients identified as not altered with the short CGA, there is no patient (0%) with a CFS score higher than 3. Thus, the short CGA does not produce a false negative with a CFS score higher than 3. Among patients with one altered score and patients with two or more altered scores, we capture a higher percentage of HIV patients with a CFS score higher than 3: 16% and 63%, respectively. Note that there are few patients left in the Controls group who have CFS available, so it’s difficult to assess this comparison for this group. We have included Table S4, which describes CFS based on short-assessment values. The table's results have been included in a new section, 3.4, of the manuscript text.
There is a very low percentage of patients with altered FP among non-altered patients assessed by the short CGA (0% in Controls and 4% in HIV). This percentage increases to 6% in Controls and 11% in VIH for subjects with one altered score and to 25% in Controls and 27% in VIH for subjects with two or more altered scores in short CGA. You can see this comparison in Table S3.

Comments 11. There is no doubt that assessing IADLs, lower extremity function with the SPPB, sleep hygiene, and cognition and functional status are all important. What is not clear is how the authors recommend using these 5 parameters. Only 50% of the PWH who had >4 abnormal tests on the original “complete” CGA had more >2 abnormal results on the short CGA. What do the authors suggest should be minimum criteria to refer for a full CGA?

Response 11: This is an excellent suggestion. We propose as a preliminary criterion the presence of ≥2 abnormal results in the short CGA to consider a full CGA, although this will require future validation.
It would only be 50% of those with more than 2 abnormal results on the short CGA who have more than 4 abnormal tests on the original complete CGA, which is not the same at all. According to what is stated, it would be 72% (42/42 + 13 + 3) of patients who have more than 4 abnormal tests on the original CGA that have more than 2 abnormal results on the short CGA, which is a significantly higher percentage.

Comments 12. At the very least a comparison between the short CGA (how many abnormal?) and an already validated and recommended frailty metric should be considered. The authors have already assessed 2 such metrics, the CFA and the FP (although the FP does take about 10-15 minutes to complete and requires the availability of an expensive hand-grip dynamometer). Determining the CFS takes less than 10 minutes and no equipment.

Response 12: In the supplementary material, we have included a descriptive table comparing the percentage of each CFS score for each category of the Short CGA (no abnormal results, one abnormal test result, and two or more abnormal test results). Also, the comparison with the FP metric was already present in Table S3. Examining these descriptive tables, we can see that the short CGA aligns with these two metrics in that it yields a higher percentage of high score values when altered.
There are some advantages to using the short CGA. The short CGA takes into account many more dimensions beyond frailty. Therefore, it is a much more comprehensive score, without incorporating all the dimensions of frailty, which would make it very costly to implement.

Reviewer 2 Report

Comments and Suggestions for Authors

The paper addresses the data mining and dimension reduction in a comprehensive - 65 geriatric assessment (CGA). The authors seek to develop a short form of the assessment based on method of principle components (PCA).

They claimed the following main results in Conclusion:

This study shows the most effective assessments for PWH aged ≥50 were the Lawton 215 IADL scale, SPPB, Barber questionnaire, PSQI, and EACS cognitive screening questions'.

However, it is not explained how these short assessment protocols are related to the leading principle components of PCA. What is the correlation between each of these statistics and the first component

of PCA for the full dataset? Why the principle component of PCA cannot be used itself for the assessment?

In my view, the detailed statistical analysis would benefit this study.

Author Response

Comments 1:
The paper addresses the data mining and dimension reduction in a comprehensive geriatric assessment (CGA). The authors seek to develop a short form of the assessment based on method of principle components (PCA). They claimed the following main results in Conclusion: This study shows the most effective assessments for PWH aged ≥50 were the Lawton IADL scale, SPPB, Barber questionnaire, PSQI, and EACS cognitive screening questions'. However, it is not explained how these short assessment protocols are related to the leading principle components of PCA. What is the correlation between each of these statistics and the first component of PCA for the full dataset? Why the principle component of PCA cannot be used itself for the assessment? In my view, the detailed statistical analysis would benefit this study.

Response 1:
We appreciate the reviewer's insightful comments and the opportunity to clarify the relationship between the selected short CGA tests and the principal components derived from PCA. 1. **Correlation with Principal Components:** We selected the short CGA tests (Lawton IADL, SPPB, Barber questionnaire, PSQI, and EACS cognitive screening questions) because they most effectively captured the two underlying dimensions of variability present in the full CGA dataset. This selection was guided by the factor map from the PCA, which visualizes the correlation of each variable with the first two principal components. In this map, the x-coordinate represents the correlation with the first principal component (PC1), while the y-coordinate corresponds to the correlation with the second principal component (PC2).
Additionally, the vectors are color-coded to reflect each variable’s contribution to the total variance explained by the principal components. We prioritized tests that had strong correlations and high contributions, especially those aligned along distinct directions, indicating their unique explanatory value. These selected tests were chosen to best reconstruct the two principal dimensions identified by the PCA. The following table presents the correlation and contribution values for all variables across the two PCA analyses performed:

Table on the PDF

From both PCAs, we can see that Lawton IADL, SPPB, Barber questionnaire, PSQI, and EACS cognitive screening questions have higher correlation values and contribution. Although Barthel also has high values in the numeric PCA, Figure A.1 shows that it points in the same direction as the Barber questionnaire, which has more correlation and contribution. The Barber questionnaire is also more important when we look at the PCA alterated. Conversely, despite being highly correlated with the Pittsburg score in the categorical PCA, the EACS cognitive screening questions was included to have at least one score that may capture the participants’ cognitive dimension as explained in the manuscript.
2. **Justification for Not Using PCA Scores Directly:** The principal components derived from Principal Component Analysis (PCA) are calculated as linear combinations of all the original scores. This means that, in order to compute the principal components for any new data point, one must first obtain measurements for all of the original scores. Consequently, while PCA effectively reduces the dimensionality of the data it does not eliminate the need to collect the full set of original scores in the first place. Instead, PCA primarily helps identify which scores account for most of the variability in the dataset highlighting the most informative scores.
We hope this explanation addresses the reviewer's concerns and clarifies the rationale behind our methodological choices. We will include this detailed statistical analysis in the revised manuscript to enhance the clarity and robustness of our findings.

Round 2

Reviewer 1 Report

Comments and Suggestions for Authors

I thank the authors for their comprehensive and informative replies to my comment, 

I have no further concerns.